# Mass media campaigns and the 'file drawer problem': A mixed methods study of how to avoid campaign failure

**James Kite** [1]*, **Margaret Thomas** [1], **Bill Bellew**[1], **Adrian Bauman**[1], **Anne Grunseit**[1,2]

**1** Prevention Research Collaboration, Sydney School of Public Health and Charles Perkins Centre, University of Sydney, Sydney, NSW, Australia, **2** School of Public Health, Faculty of Health, University of Technology Sydney, Ultimo, NSW, Australia

* james.kite@sydney.edu.au

**Data Availability Statement:** Data cannot be shared publicly to protect participant confidentiality. Specifically, the raw data are identifiable as participants described their roles in

## Abstract

Mass media campaigns are frequently used to address public health issues. Considering the considerable cost, there has been little analysis of why campaigns sometimes fail. This study used a sequential mixed methods approach to explore the mechanisms that can lead to failure and to identify what can be done to avoid or overcome common mistakes in campaign planning, implementation, and evaluation. We conducted interviews and a survey with 28 public health social marketing and mass media campaign experts over three rounds of research and analysed the data thematically, generating themes inductively. We identified four systemic factors that drive success: long-term strategic thinking and commitment, understanding the campaign context, doing and learning from evaluation, and fostering strong relationships. The factors did not operate in isolation, rather good (or poor) execution in one area was likely to influence performance in another. The experts also emphasised that a change of political context could drastically affect one or more of the identified factors. Our analysis showed that campaign failures are not simply individuals making mistakes. Systemic issues throughout the planning, execution, and evaluation phases need to be addressed if campaign outcomes are to improve.

## Introduction

Mass media campaigns (hereafter 'campaigns') are a common strategy for addressing public health issues. These campaigns feature a communications initiative, disseminated via mass media (which includes 'old' media, such as television, radio, and billboards, as well as 'new' media, such as digital banner advertising and social media), that aims to build knowledge, shift attitudes, and/or encourage behaviour change [1]. They are sometimes part of broader social marketing programs and/or complemented by other initiatives. Examples of these initiatives include environmental or policy changes, or community-based programs; this approach has seen considerable success in areas such as tobacco control and road safety [2, 3]. Despite their popularity, what constitutes 'success' and 'failure' in campaigns and the factors that contribute to whether they succeed or fail remain elusive and open to debate.

detail and referenced specific campaigns that they had worked on. Additionally, participants did not consent to sharing their interview transcripts. Data access requests could be directed to The University of Sydney's Human Research Ethics Committee on +61 2 8627 8176 or email human.ethics@sydney.edu.au, citing project number 2020/091.

**Funding:** The author(s) received no specific funding for this work.

**Competing interests:** The authors have declared that no competing interests exist.

While criteria for what constitutes a social marketing campaign have been proposed [4–6] and guidelines on the best practices for mass media campaigns have been set out [7–12], these fail to incorporate contextual factors that influence how campaigns are designed, implemented, and evaluated. Further, the discourse on campaigns in the published literature has largely centred on factors that lead to success, with little attention paid to failure or what can go wrong. There may be several reasons for the dearth of published studies on campaign failures. Firstly, there may be an assumption that we can understand campaign failures as being the result of the lack of critical success factors. However, without explicitly examining failures, we cannot be certain this is the case. Secondly, evaluations are often too narrow in scope, under resourced, or, worse still, not conducted at all [13–15]. Finally, there is a lack of transparency whereby evaluation findings may be kept 'for internal use only', especially if they show little or no impact or negative impact. This is a problem experienced not only in campaign evaluation but in intervention research more generally (what Rosenthal [16] dubbed the 'file drawer problem'). Consequently, there is a lack of evidence that can be used to explore the circumstances under which campaigns succeed or fail and a lack of consensus on what constitutes success and failure in campaigns.

Recently, researchers have begun to address this gap in the evidence base for social marketing campaigns more broadly. Through interviews and surveys with social marketing practitioners, their studies have identified factors that led to social marketing campaign failure [15, 17, 18]. These include a lack of formative evaluation and research, ad hoc approaches to planning, a failure to use theories and frameworks to underpin campaign design, a narrow or exclusive focus on downstream factors, and failing to develop a common vision with stakeholders. One study of social marketing campaigns in Aotearoa New Zealand highlighted that external influences (such as funding cuts, political ideologies, and interference) can also lead to campaign failure [19]. However, this growing evidence base has some limitations as the samples were skewed towards those with experience in design and implementation of social marketing programs, and less commonly included participants who had extensive experience in evaluation or as consultants. The perspective of these groups is valuable because they play critical roles in the design, implementation, evaluation, and reporting of campaigns. Further, these studies focus on the use of social marketing for any issue. Given the widespread use of mass media in public health [12, 20], a detailed analysis of this campaign context would be informative for policymakers' and administrators' decision-making.

As in any field, it is important to share negative and null results from prevention campaigns so that the community can learn from mistakes and improve practice [16]. To address the file drawer problem, we sought the views of those with extensive campaign experience about what can go wrong and how these mistakes could be avoided. Specifically, we asked public health social marketing and mass media campaign experts:

1. What are the components and/or processes for mass media communication campaign planning and implementation that are most at risk of poor execution and why?

2. Are the current measures of campaign success appropriate?

3. Can common mistakes in campaign planning, implementation, and evaluation be avoided and how?

Our study aimed to improve understanding of the mechanisms that lead to the failure of health-focussed mass communication campaigns and generate recommendations and solutions. To say, for example, that lack of formative research is important and therefore campaigners should do more formative research begs the question of *why* the formative research was *not* done. By taking an iterative and analytical approach to examining the systemic,

political, and contextual factors that contribute to unsuccessful campaigns we have taken this research to a level not previously examined. Our study provides new evidence about how criteria for 'success' may contribute to campaign results being relegated to the file drawer in the first place and how, on the other hand, various factors that may be difficult to change will continue to contribute to campaign failures.

## Methods

Our study used a sequential mixed methods approach to data collection that involved three rounds of research with experts in social marketing and mass media campaigns. The study was approved by the University of Sydney Human Research Ethics Committee (Project No.: 2020/091). All participants provided written consent.

### Recruitment

'Experts' included researchers and practitioners, including consultants, from within public health who had published regularly on campaigns and/or had an extensive career in planning, implementing, and/or evaluating campaigns. In this way, our study addresses the skewed sample problem of earlier studies by having a broader definition of 'expert'. Participants were recruited via email between 10 March 2020 and 7 May 2020 and asked to participate in two rounds of research (either Round One and Round Two or Round Two and Round Three; Fig 1). We developed our initial list of experts through purposive sampling, identifying authors of campaign evaluations and commentaries and drawing from our own network of international contacts. We used a purposive sampling strategy, aiming to recruit a range of views and experiences. We then used passive snowballing to identify additional experts; participants in Rounds One and Two were asked to pass on the study details to any experts they thought would be eligible for this research. We aimed to have a mix of different types of experts in Rounds One and Three to ensure different perspectives on campaigns were heard at each stage of the research.

### Data collection and analysis

Rounds were structured sequentially so that Round One focused on identifying common points of failure in campaigns through qualitative interviews, Round Two on assessing which of the points of failure were the most significant in terms of their potential impact on campaign effectiveness using a quantitative survey, and Round Three on identifying ways of solving or avoiding the most significant points of failure with further qualitative interviews. Round One comprised semi-structured interviews of approximately 30 minutes to an hour, conducted via a videoconferencing platform with the lead author. The interviews were exploratory in nature, covering the participants' experience with campaign failures, their opinions on common mistakes in campaigns, the reasons for them, and how they could be avoided or mitigated. Interviews were audio recorded and transcribed verbatim.

We conducted an iterative thematic analysis [21] of these interviews, supported by NVivo 11, to identify points of campaign failure. Themes were inductively generated from the interview material by JK after listening to the recordings and reviewing the transcriptions for all interviews. Initially, JK developed a code frame based on how participants characterised their experiences with campaigns and their definitions of campaign success and failure. The code frame was discussed and modified with MG and AT following their own review of a subsample of transcriptions. JK then used the revised code frame to generate initial themes that focused on points in the campaign process at which failures could occur and the reasons for these. The initial themes were discussed with MG and AT and used to inform Round Two.

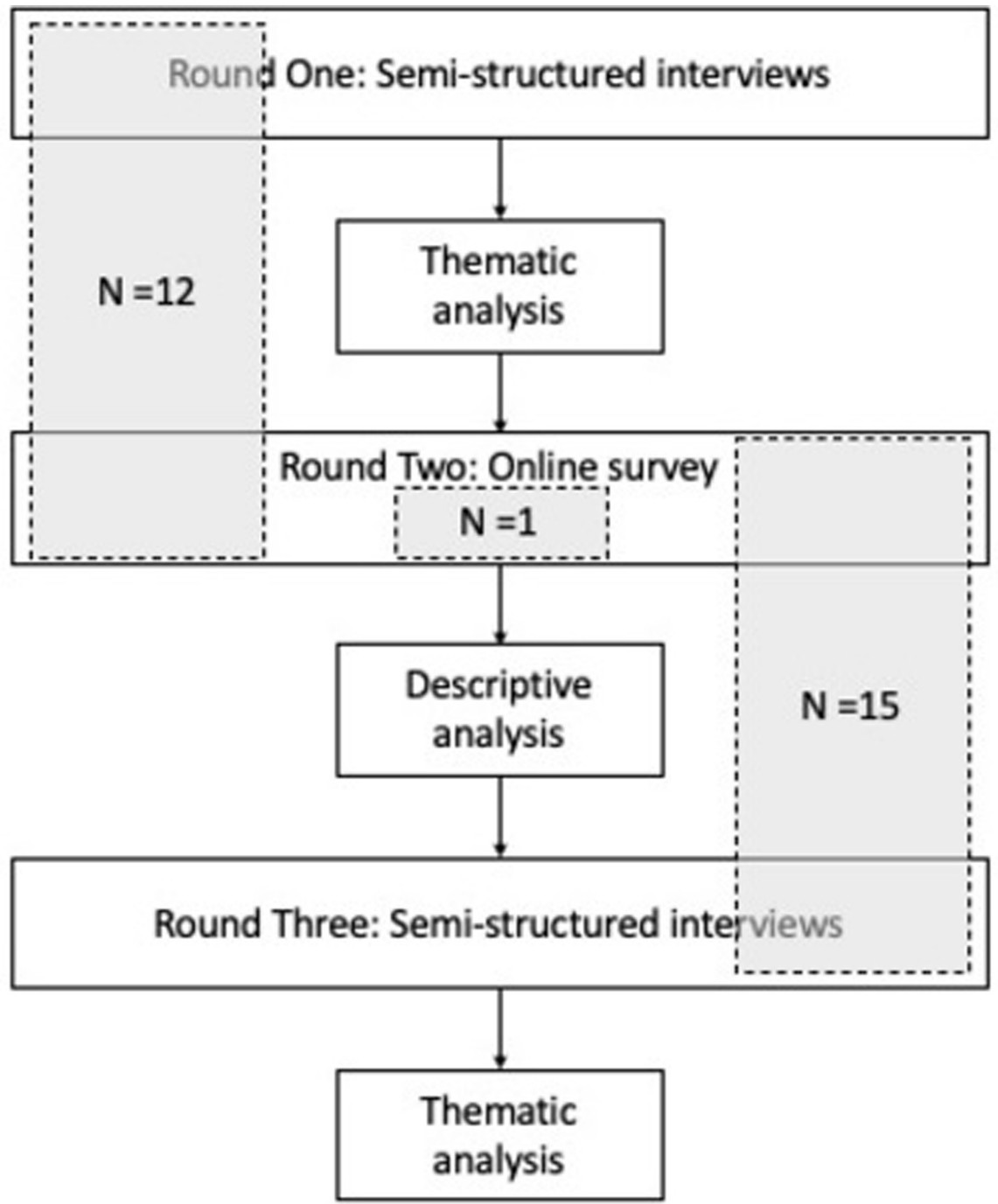

**Fig 1. Study design flow chart.**

Using the points of campaign failure and subcategories of failure from Round One, we prepared a structured online survey for Round Two, which was sent to all participants. The aim of Round Two was to test the failures we had identified in Round One and their relationship with one another. Participants were asked to consider the results of Round One, including whether they felt the results reflected their experience and whether any important elements had been

missed. Participants were also asked to consider the significance of the failures, the frequency with which the failures occurred, and whether the failures were avoidable. The survey included demographic questions on campaign role(s), where we asked participants to classify themselves as a researcher/academic (i.e. those with responsibility for research and/or evaluation of campaigns), 'pracademic' (i.e. those with responsibility for both campaign design and delivery and research and evaluation of campaigns) [22], or practitioner (i.e. those with responsibility for campaign design and delivery) and their level of experience with campaigns, as measured by the number of years they had worked in the field.

We summarised the questionnaire results descriptively and combined them with the results from Round One to a generate results summary that was shared with participants for Round Three (see S1 File). The Round Three interviews were conducted in the same manner as Round One, except that these interviews explored the accuracy, relevance, and usefulness of the results summary, any issues raised in the survey results, and how the identified failures could be avoided. Analysis was thematic, with JK reviewing and revising the initial themes based on the same immersion process described for the Round One interviews. MG and AT were consulted on the revised themes and the final themes were developed based on discussion and consensus. We focused particularly on themes that could elucidate solutions to the points of failure identified in earlier rounds.

Copies of the Round One and Three discussion guides and Round Two questionnaire are provided in S2–S4 Files.

## Mixed methods integration

This study was a mixed methods sequential design, integrating the methods, as described above, and the data at the analysis and interpretation stages [23]. The qualitative results from Rounds One were used to identify themes, while the quantitative results from Round Two were used to explore the relative importance and perceived impact of the themes on campaign success and failure. We gave more weight to the Round Three interview data [24], revising our analysis following these interviews and generating the final conclusions using the Round One and Three interview material combined and integrated with the Round Two survey results. Further, after initial analysis of Round Three, we determined that similar themes were emerging across all rounds of the research. The strength of our study is that we designed the three rounds of data collection as a cohesive investigation and report the results from all rounds together, rather than separately.

## Results

### Description of the sample

We invited 38 experts to be part of the study, with 28 agreeing (12 participants completed Round One and Round Two, 15 completed Round Two and Round Three, and one participated only in Round Two). From the survey, most participants (n = 19) had more than 15 years' experience in social marketing and mass media campaigns, while the remainder had between 5- and 15-years' experience. Most worked predominantly in high income settings, most commonly Australia but also including the United States, the United Kingdom, Ireland, and Western Europe., with only four working predominantly in low- or middle-income settings that participants had experience in, including countries in south-east Asia, the Pacific Islands, and Africa. Ten participants characterised their role as being a researcher/academic, 11 as a 'pracademic' (someone who does both campaign design and delivery and research and evaluation of campaigns), and seven as a practitioner. In the interviews, participants described their roles in campaigns more specifically as including strategic planning, campaign design,

appointing and coordinating specialist services (e.g. creative agencies, research agencies, media planners/buyers), provision of expert advice, developing and producing campaign materials, conducting formative, process, and/or impact/outcome evaluation, and analysis and reporting of evaluation results. Participants had worked on campaigns targeting a wide variety of health issues, including alcohol consumption, road safety, tobacco and vaping control, physical activity, nutrition, immunisation, and cancer screening.

## Round One and Round Two summary

Based on the Round One interviews, we generated three overarching categories of failure: strategic failures (relating to the decisions to conduct a campaign and how it relates to other initiatives), process failures (relating to how the campaign is managed from conception through evaluation), and implementation failures (relating to the creative design and implementation of the campaign). Each of these categories of failure had subcategories (described in Table 1). Experts generally described all failure types as having a significant impact on the potential success of a campaign, but equally almost all were felt to be highly amenable to change.

While the categories and subcategories of failure received broad endorsement as being appropriate classifications from participants in Rounds Two and Three, Round Three participants described the relationship between the failures as a more complex and nuanced process than what we had represented in our results summary:

*For me, there's a lot more [going on] in my experience of the way it happens. [Round Three, Participant 1, Practitioner]*

Following review of the initial themes, the four themes developed following Round Three were more solution-focussed than the initial categories and subcategories of failure. Consequently, they highlight the factors that participants felt were critical for avoiding campaign failure: long-term strategic thinking and commitment; understanding the campaign context; doing and learning from evaluation; and fostering strong relationships (Fig 2).

## Long-term strategic thinking and commitment

Participants strongly endorsed the need for a long-term focus on campaigning, arguing that short-term approaches were at best ineffective and a waste of resources. The Round Two survey results underscored this, with about half of all participants ranking isolated campaigning as one of the most significant failure types in terms of its impact on campaign effectiveness and one of the most frequent points of failure. However, it was also ranked as the most amenable to change among the strategic failures.

Participants felt that adopting a long-term focus meant planning several years ahead and being ready to capitalise on favourable changes in the context in which campaigns take place, especially the political environment. It also meant committing campaign resources to achieve meaningful change:

*Time, I think is vital here, you've got to be in this for the long run. [Coca-Cola have] been at this job for 100 years trying to get this right. They make enormous cockups at times [but] they had an idea of their destination. I sometimes doubt that public health has that. [Round One, Participant 3, Researcher]*

*In the short term, you can't change the system. But in the long term, you can. If you look at [road safety]–which goes back to consumer-focused work in 1989, to get people to slow down and don't drink and drive, wear your safety belts, all that sort of stuff–it's now the [transport*

**Table 1. Description of subcategories of failure from Round 1 and Round 2.**

| Failure type | Description |
|---|---|
| **Strategic failures** | |
| Isolated campaigning | Campaigns are conducted in the absence of or in isolation from a larger strategy to address the health issue. |
| Short-termism | Campaigns are conducted over too short a time period or at insufficient intensity to have a meaningful, sustained impact on the health issue. |
| Political or bureaucratic interference | Campaigns are negatively influenced by political or bureaucratic pressures, such as by weakening or avoiding particular messages or removing funding. |
| Lack of adequate funding or resources to execute the campaign strategy | The funding and resources provided to campaign implementation are not sufficient to achieve the campaign's objectives. |
| Risk aversion | A tendency to avoid campaign messaging that will attract opposition, particularly from political opponents or private-sector industries, and instead adopt 'safe' campaign messaging that focuses on individual behaviour change. |
| **Process failures** | |
| No or poor formative research | Campaigns that lack adequate formative research to inform their design, implementation, and/or evaluation. |
| Weak relationships between funders, researchers, and creatives | Undervaluing the relationships that develop between funders, researchers, and creatives or impeding the development of such relationships through bureaucratic processes. |
| Not learning from past campaigns (own or others) | Not conducting adequate process, impact, and/or outcome evaluations of campaigns. It also extends to not reporting the results of evaluations or reporting evaluations in a way that over-emphasises campaign achievements and ignores or downplays failures or weaknesses. |
| Inappropriate or poor processes for the selection of creative agencies | The tendency to require creative agencies to prepare campaign ideas in a competitive pitch process. Such processes result in ill-informed or 'safe' campaign ideas as creative agencies work in isolation to develop their ideas and focus on what they think will get them the contract and not on what would be best for campaign outcomes. |
| Inappropriate or poor process for campaign approval | Slow or unresponsive processes for approval of campaigns that discourage innovation and adaptability in campaigns. |
| **Implementation failures** | |
| Inappropriate or poor measures of campaign success | Setting or focusing on measures of campaign success that do not (adequately) reflect the campaign objectives ("measurement failure"). |
| Inappropriate or poor communication channels | Selecting communication channels that are not appropriate for the campaign audience ["delivery failure"]. |
| Inappropriate or poor campaign objectives | Setting vague, unclear, or unrealistic objectives or objectives that do not reflect the causes of the health issue being addressed. |
| Inappropriate or poor creative/messaging | Developing campaign messages that do not have the desired impact on the target audience and may have negative consequences such as demotivation or stigmatisation ["message failure"]. |

*department that] are working with governments and car manufacturers getting them to make safer cars and safer roads. So that's the kind of length of tenure that you need for systemic change. And that can be problematic in a political system. [Round Three, Participant 8, Researcher]*

The need to adopt a 'systems thinking' approach [25] came through strongly from some participants, especially in Round Three. They emphasised the need to see health issues as a system and to consider what role a campaign should play in achieving systems change:

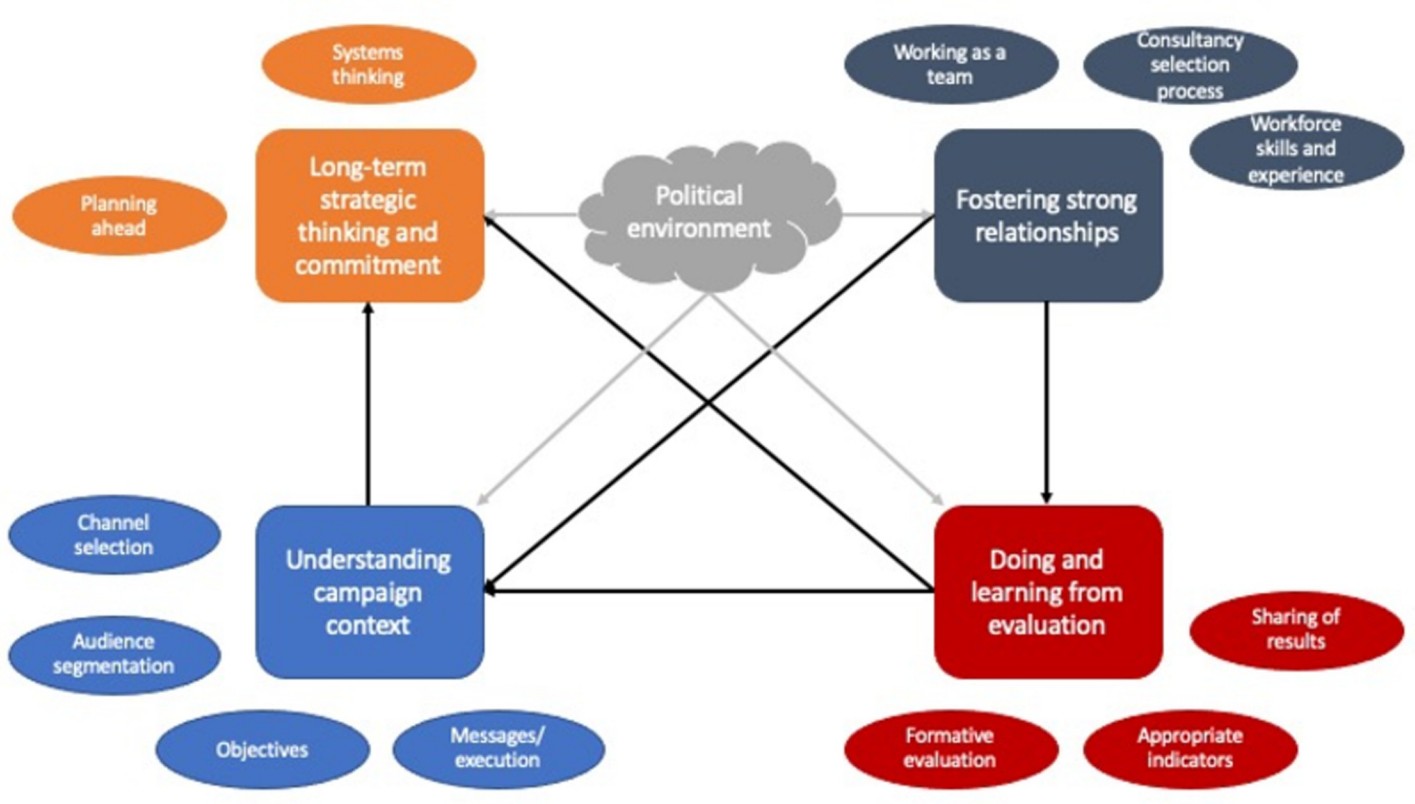

**Fig 2. Conceptual map of findings.**

*[Campaigners should] start thinking more systemically, start approaching things from a process perspective, start joining up the dots, start working in partnerships, and get beyond simplified decision-making and start to deal with the complexity that's there. Understand the whole system, and not just the part that you are working in. [Round Three, Participant 4, Researcher]*

It was evident across all rounds that participants felt that campaigns were often treated separately from broader strategies designed to address a health problem. Campaign isolation was described as a key contributor to failure, and is perpetuated because campaigns are less likely to face public and industry opposition compared with more 'intrusive' measures:

*Ministers do want to be seen to be doing something, [but] they want to get into a space which is uncontroversial. "We're going to run a nice campaign over here and that means we don't have to do anything about the [alcohol industry's] behaviour." This is enticing because it is a quick fix. [Round One, Participant 3, Researcher]*

### Understanding the campaign context

Participants made it clear that understanding the target audience and what objectives, channels, and messaging were appropriate for them was critical for campaign success. Poor

objectives were felt to be particularly damaging for campaigns, with participants in Round Two ranking this as among the most frequent and impactful failure types. Key points of failure for objectives were that they lacked ambition, target the wrong aspects of a health problem, or are poorly constructed:

*I think objectives often can be too vague and not specific enough, and so then when you get your evaluation back, you can't actually tell if you've had an impact on it. [Round One, Participant 2, Practitioner]*

*Once social media and other types of digital media were introduced, it became a lot harder to set objectives because often there was no evidence guiding where we were setting those objectives. [Round Three, Participant 11, Researcher]*

Understanding the campaign context also meant knowing which messages would resonate with the target audience and what channels should be used for those messages. Some interviewees spoke of campaign materials that were created to gain acclaim, rather than to achieve public health outcomes, and how these invariably failed to achieve those outcomes:

*[That] campaign didn't really fit the objective of what we're aiming to do. I think that can be a problem and I think if people do a campaign just because they think it's a good idea or it's a clever idea, that can be frustrating. You have to pull back and say, "This is our target audience this is what we're trying to achieve. We're not trying to win [the creative agency] an award." [Round Three, Participant 12, Researcher]*

Others highlighted the need to carefully select communication channels that were appropriate for the target audience and the resources of the campaign. One participant saw the intersection of channel and audience as particularly pertinent in a time when the communication landscape had changed significantly:

*In Australia, I don't think we will ever see the big TV-led campaigns that we used to have. People just don't watch TV like they used to. So, it is a challenge to try and achieve that reach in the population using [new] media. [Round One, Participant 5, Pracademic]*

Participants were, however, optimistic that failures to take campaign context into account were avoidable, ranking all the subcategories as highly amenable to change in Round Two. They also believed that these failures were strongly associated with a lack of skills and experience in the campaign team, isolated campaigning, and a lack of formative research.

## Doing and learning from evaluation

Evaluation was felt by many participants to be critical yet often overlooked or poorly done. Participants mentioned this in relation to all levels of evaluation, from formative through to outcome evaluation. The lack of or poorly conducted formative research was ranked as the most significant process failure in terms of its impact on campaign effectiveness in Round Two. Interviewees described it as one of the biggest contributors to campaign failure:

*When you put a campaign on air, it's an expensive business, you get one opportunity to get it right, it's too late then to be ticking boxes and doing testing and having hindsight about whether it was a good idea or not. I mean, there are campaigns that have been pulled or*

*which haven't worked, but I would argue that they could have been avoided if the formative work had been done properly. [Round One, Participant 7, Pracademic]*

*They ended up taking [the campaign] off air because it just upset too many people. Now when I heard that happening, I [thought] "I bet they didn't do any research or do the right research before they put that on air." It was really obvious. They should've talked to [the target audience]. They could have [avoided failure with better formative research]. They should have. [Round One, Participant 5, Pracademic]*

Poor formative (planning, pre-campaign) research was ranked as highly amenable to change in Round Two. Interviewees believed that upskilling campaign teams and ensuring there was sufficient time and resources allocated to formative research would ensure it was appropriately and thoroughly conducted, thereby significantly reducing the risk of campaign failure.

Others pointed to the selection of appropriate measures or indicators of campaign success as an important consideration. Selecting inappropriate measures of success led to poor decisions in campaign design and implementation, and was particularly a risk with digital campaigns, where the number of potential indicators could be overwhelming, confusing, or lead to erroneous conclusions:

*That's still something that we're educating stakeholders on, that just because you can measure the clicks [or] landing pages, it doesn't mean to say it's the right thing to do, and sometimes that can be a major distraction. [Round Three, Participant 7, Practitioner]*

*You know what I hate? That campaign won all sorts of awards. And it was watched by millions of people, and it was spread all over the world. It was viral. Do you know what the behavioural results of it [were]? Fuck all. Nothing. It [wasn't] effective. And yet so many people hold it up there as "let's do something as good as this [because it went viral]." [Round One, Participant 6, Practitioner]*

Additionally, participants frequently lamented that those involved in campaigns did not or could not learn from previous campaigns. In part this was because of the lack of accessible reporting of evaluation results from campaigns, with one participant suggesting that a repository for campaign evaluations may help alleviate this:

*If we had a resource that was available for the newbies to tap into. . . then we could be learning. But it can't be embedded within the specific organizations. It has to be above that, and to be readily available. [Round Three, Participant 8, Researcher]*

Others spoke of barriers to learning from evaluations, including nuancing publications to hide or downplay politically sensitive results, as well as a lack of capacity to produce an academic paper, competing priorities, and reluctance to publish null or negative results:

*In my time [in a government role], we did publish a little, but our capacity wasn't there to publish and share as much as we would have liked to. [Round One, Participant 11, Practitioner]*

*If I had to guess, I would guess that [campaign evaluation] probably will never be published. The incentives start to go away. You move on to other projects and you get busy and you're not funded to work on that project anymore. So, you really have to find the time somehow to do it. [Round Three, Participant 6, Researcher]*

*What I see is that all publications are about, "Hey, we have been really successful." All papers try to find out the way to say that they'd been very successful. There's no learning from that. [Round One, Participant 8, Researcher]*

### Fostering strong relationships

Several interviewees described cultural and structural barriers to effective campaign practices. Among these, strong relationships between all partners involved in campaign design, implementation, and evaluation were seen as key to campaign success, although this was ranked lower in Round Two than other failure types in terms of their impact on campaign effectiveness. Some interviewees felt the value of relationships was downplayed, forgotten, or ignored by campaign processes. Consequently, participants felt that campaigns are often delivered in silos, with partners from different industries or disciplines (e.g. government departments with research and/or creative agencies) managed independently by the funder, rather than as a team. Campaigns were felt to be most likely to succeed when all parties were able to work in partnership or as a team to plan, deliver, and evaluate the campaign:

*There's part of me that likes the idea that it becomes a relationship between the policy makers and implementers [with] the creative minds [and] the planners [and] the evaluation people, so you've got them all in the room at the same time. [Round One, Participant 9, Pracademic]*

One significant barrier to effective relationships described by participants was the processes for selection and appointment of campaign specialists. This included creative agencies and media strategists, as well as evaluators, especially those conducting formative evaluations on behalf of clients. Participants felt that the governance structures for these appointments put too much emphasis on probity and not enough on the value of established relationships. Selection processes for creative agencies were seen as particularly problematic when the funders selected the agency based only on a competitive pitch process:

*Giving [creative agencies] a briefing document, a 20-minute briefing on the campaign, and then expecting them to go and work in a silo to come back with this amazing pitch idea that they wow you with is just so unrealistic... [Round One, Participant 2, Pracademic]*

Such approaches, it was felt, encouraged 'safe' ideas from the creative agencies who deliver what they believed would be acceptable to the funding agency, as opposed to what would be effective for the target population. Similarly, the fact that these specialists are often contracted to work on a per campaign basis, rather than ongoing, meant it is difficult for relationships to develop and to learn from experience.

Some participants spoke of the need to build the skills of the campaign workforce, with mentoring suggested as one avenue to facilitate improvement. Others suggested that those with campaign skills and experience should seek to establish networks across public health issues to share their knowledge. These participants felt currently it was more common to network within a single public health issue and that consequently the pool of knowledge and experiences relating to campaigns was limited. Those with experience working in low- and middle-income settings also highlighted that different settings require different approaches to building the campaign workforce as the base-level of understanding varies:

*[In some the countries I have worked in] the idea of making a 30 second TV spot and broadcasting it is a very unusual thing to do... Even though they're used to seeing ads for all sorts of*

*products on their TV, the idea of using similar marketing strategies to influence something like health behaviours, that can be brand new. And that can be a challenge. [Round Three, Participant 1, Practitioner]*

## The way forward

When asked about whether campaign failure could be avoided and how the systems and processes that support campaign development, implementation and evaluation could be improved, participants were almost universally optimistic that change was possible. In the Round Two results, for example, when asked about amenity to change the identified failures, for almost all the failure types, a clear majority (above two-thirds) of participants felt they were amenable to change. Round Three participants confirmed that optimism:

*There has to be hope. If there isn't hope, it would be a very dark day. I think the question isn't "are [the failure types] amenable to change?" I think the question is the timeframe to that change. [Round Three, Participant 4, Researcher]*

While the four factors were important for understanding how to improve campaigns on their own merits, they also interacted with one another. Participants described this as a symbiotic relationship, with good (or poor) processes in one area felt to greatly increase the likelihood of good (or poor) performance in another:

*It's like, if you make a process failure, it's probably also going to roll into implementation and strategic failures, right? That they all build on each other. [Round Three, Participant 2, Researcher]*

Additionally, all four factors were influenced by the political environment of a campaign. This primarily seemed to amount to interference, with participants describing it as authority figures (often politicians or senior executives of government and non-government agencies) exerting influence to shape a new campaign or alter or cancel an existing one. Such influence was viewed as inappropriate and/or damaging to the potential effectiveness of the campaign:

*That happens often in government as well, where going through the various levels of approval processes, a campaign creative will get watered down, and so it just doesn't have the same cut through or the same impact with the audience. [Round One, Participant 2, Practitioner]*

*If you're a policy maker coming up with [a] campaign and you need that signed off by the government, then you're [more likely to get an individual behaviour change campaign] signed off. A cynic might say, "I will sign anything off that we know doesn't work." Anything that's contentious or really hard hitting, "Well, hang on a minute." [Round One, Participant 1, Researcher]*

Yet, unlike the other factors described above, participants largely felt that such interference was almost unavoidable; in Round Two, political interference was one of only two failure types that most participants (68%) felt it would be difficult to change (the other was lack of funding/resources at 71%). According to our participants, this reinforced the need for those involved in campaigns to be ready to capitalise on disruptions in the political environment, such as a change in government.

However, some participants recommended separating campaign decision making from the political cycle where possible. For example, one participant spoke of a government agency providing long-term funding to non-government organisations to run campaigns on their behalf. While she acknowledged that this model relied on the availability of suitable organisations and individuals, creating this distance meant that the relatively frequent changes in the governing party or the politician in charge were less likely to affect campaigns:

*I think there's a lot of benefits with the model that we've got, which is outsourced, in that it helps to quarantine the money because the money is a contract and you've got to honour the contract. And the contracts are normally five to seven years, at least. And there's an ability to have much harder hitting campaigns I think than is necessarily possible these days for government delivered campaigns. [Round Three, Participant 5, Practitioner]*

Most notably, though, having a culture and system that supported strong relationships between campaign stakeholders, including those in the political realm, seemed to underpin success in understanding the campaign context and in evaluation (Fig 2). These in turn would increase the likelihood of adopting long-term strategic thinking, including being ready to capitalise on favourable changes in the political environment. Bringing about change, then, requires creation of an environment that supports the adoption of more appropriate campaign practices as identified by our participants.

## Discussion

Our study aimed to open the file drawer on mass media campaigns and find out what we can learn from its contents. It contributes to a burgeoning literature on failures in mass media communication and social marketing campaigns such as the lack of formative research and failure to plan appropriately, as well as the potentially negative influence of political ideologies and interference [15, 17–19]. Our findings reinforce the importance of formative research, selection of appropriate measures of success, and long-term strategic and systems thinking. They also support mass media communication and social marketing campaign best practice by emphasising the need to understand the campaign context, including the target audience [6]. However, where we have extended previous research is by showing that campaign failures are not simply due to individual campaigners or campaign teams making isolated mistakes, but rather that the environment in which campaigns are developed, implemented, and evaluated is often not conducive to the use of best practice.

One key element requiring change is the structures and culture of campaign governance, something that has not been highlighted in previous research. Currently, governance does not support communication and relationship building, and our participants felt that these are critical in maximising the chances of success. Moreover, the strength of the relationships between campaign stakeholders exerts a significant influence on the likelihood of these stakeholders understanding the campaign context and 'doing and learning' from evaluation. Working collaboratively or in partnership is an important part of social marketing as a discipline, but often in the context of working across sectors or agencies [26]. Our findings show it is also important to consider the value of stakeholder relationships to the success of campaigns. Competitive pitch processes for selecting creative agencies, for example, was one practice that our participants highlighted as needing to change. While safeguards to ensure the appropriate use of funds are necessary, it appears that the pendulum has swung too far towards probity concerns, with no consideration given to the harm pitch processes can cause to relationships, or the risk it poses to sound definitions of campaign success. We also found that governance structures tend to allow

for political interference, an issue that has been highlighted elsewhere [18]. Consequently, campaigns tend to be 'safe', targeting individual behaviour change, even where there is evidence that such approaches are potentially harmful [27–29]. Governance structures need to be reformed to value relationships and relationship-building to foster effective and sustained collaboration and allow multi-disciplinary campaign teams to develop. They also need to minimise the possibility of political interference in campaign design, implementation, and evaluation.

Another key finding not discussed in previous studies is the importance of selecting appropriate measures of success and how inappropriate measures can undermine a campaign before it has even begun. This is an issue that extends beyond just evaluation to also include understanding the campaign context and long-term strategic thinking. For example, evaluation indicators may be appropriate to the objectives, but if the objectives are narrow and target only individual behaviour change when social, environmental, or policy-related causes are more significant, then the campaign may fail to achieve meaningful, population-level improvement in health-related outcomes. Additionally, the increasing role of digital media in campaigns compounds this issue as so many metrics are now available that it can be easy to fixate on proximal measures such as the number of website hits, social media engagement (e.g., likes, comments, or shares), or other metrics that may not be good indicators of more distal and significant campaign effects such as behaviour change [30]. The implication for those working on campaign design and evaluation is to be especially mindful of the selection of appropriate measures of campaign success. This issue also highlights the interdependent relationship between the identified factors and how improving one area, such as selection of appropriate campaign measures, is likely to have flow on effects to address other areas, such as long-term strategic thinking. Equally, though, not addressing one issue may hinder progress in another area. Those involved in campaigns should be mindful of the interconnected relationship between the factors we have identified as they work to improve campaign outcomes.

Our experts recognised the need for greater sharing of 'warts-and-all' campaign evaluation findings. Many spoke of a lack of mechanisms to share findings, especially across public health issues (e.g., from tobacco control to road safety). The current reliance on publications in peer reviewed journals was thought insufficient because of capacity constraints and a tendency to present campaigns in a positive light. These flaws, along with others such as the slowness of the publication process and access barriers to academic publication, are widely recognised [31, 32]. Continuing barriers to sharing evaluations make it less likely that we can learn from campaign failures, but also reduce the likelihood of being able to learn from successes as well. While there are opportunities to work within the publishing system, such as engaging university partners to conduct evaluations and lead publications, this does not resolve all the obstacles inherent in the current system. Instead, a central repository, as suggested by one of our participants, could be one avenue for reducing barriers to sharing campaign evaluations across health issues, like the SNAP-Ed Toolkit for obesity prevention interventions (https://snapedtoolkit.org/). Such a repository would need to be quality-controlled and managed independently of campaign stakeholders to reduce conflicts of interest and the likelihood of funding being withdrawn. There would also need to be few barriers to being able to share information via this repository, such as requiring peer review or paywalled access. Campaign practitioners and researchers should explore establishing a repository of this kind. Doing so may also help improve awareness of mass media communication and social marketing as disciplines [33].

A strength of this study is that it looked specifically at mass media communications, as opposed to social marketing more broadly, which is important given the widespread use of mass media in public health [12, 20]. The mixed methods approach allowed for an in-depth exploration of issues and for us to test our understandings of the data from earlier rounds with the experts. Additionally, our participants also represented a wider variety of experiences and

roles in campaigns compared with previous studies. However, our sample was skewed towards those who work in high-income settings, especially Australia. It is possible that our findings may have been different had we included more people with experience in low- and middle-income and non-English-speaking settings.

Our study has reinforced and expanded on existing research on factors that lead to campaign failure by being solutions-focused and identifying how these factors could be tackled to reduce the likelihood of campaign failure. What is clear is that campaign failures are not simply a matter of inexperience or mistakes by campaign teams, but rather reflect systemic issues. The implication of this finding for practice is that addressing these issues requires seeing the campaign design, implementation, and evaluation process as a system, rather than as discrete components. It is only by recognising how poor systems contribute to poor processes that we can hope to improve mass media campaign outcomes.

## Supporting information

**S1 File. Round Two results summary.**
(DOCX)

**S2 File. Round One interview discussion guide.**
(DOCX)

**S3 File. Round Two questionnaire.**
(DOCX)

**S4 File. Round Three interview discussion guide.**
(DOCX)

## Acknowledgments

We wish to thank all the participants for their time in completing this study.

## Author Contributions

**Conceptualization:** James Kite, Margaret Thomas, Anne Grunseit.

**Data curation:** James Kite.

**Formal analysis:** James Kite, Margaret Thomas, Anne Grunseit.

**Investigation:** James Kite.

**Methodology:** James Kite, Margaret Thomas, Bill Bellew, Adrian Bauman, Anne Grunseit.

**Project administration:** James Kite.

**Writing – original draft:** James Kite.

**Writing – review & editing:** James Kite, Margaret Thomas, Bill Bellew, Adrian Bauman, Anne Grunseit.

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
