## [Decision Letter · Decision Letter 0]

12 Feb 2024

PONE-D-23-35166Mass media campaigns and the ‘file drawer problem’: a Delphi study of how to avoid campaign failurePLOS ONE

Dear Dr. Kite,

Thank you for submitting your manuscript to PLOS ONE. After careful consideration, we feel that it has merit but does not fully meet PLOS ONE’s publication criteria as it currently stands. Therefore, we invite you to submit a revised version of the manuscript that addresses the points raised during the review process.

As a reviewer I had concerns about how the methodology was described (see my review below), but with my hat on as Academic Editor, my view is that if it is going to be presented as a Delphi study, then the case for this needs to be clearly made in the paper, and the modifications need to be justified in methodological terms. This will also require additional detail in the methods section and the results section that are typically expected in Delphi studies (as summarised in the checklists that are available for Delphi studies, such as the CREDES checklist). However, on further reflection, it may be decided that the study would be better described simply as a sequential mixed methods study. This would be acceptable, and the study design would then suit the change in focus at each stage, as this would reflect a logical development of the study.

We look forward to receiving your revised manuscript.

Kind regards,

Simon White

Academic Editor

PLOS ONE

4. Please remove your figures from within your manuscript file, leaving only the individual TIFF/EPS image files, uploaded separately. These will be automatically included in the reviewers’ PDF.

Additional Editor Comments:

Academic Editor acting as Reviewer 2 (due to difficulty finding a second reviewer for this topic):

This paper concerns mass media public health campaign failure, which is an important topic, but little appears to have been published about it. As such, this paper has the potential to make a valuable contribution to the field. The paper is generally well-written, and the results feel credible, from a qualitative point of view at least, but I have concerns about how the methodology is presented.

I agree that the study adopted a sequential mixed methods design, but it used such an atypical approach to a Delphi study that it is not clear whether it can still be classified as a Delphi study. Whilst definitions vary, the Delphi technique is generally regarded as a consensus methodology in which a Panel of experts individually consider their position on aspects of a topic and in subsequent Rounds, following feedback on the collective opinion of the Panel on these aspects from the previous Round, the (remaining) Panellists reconsider their previously declared position on these aspects, towards achieving (or not achieving) consensus. No explanation or justification (e.g. using methodological literature) is provided in the paper for such deviation from even accepted modifications to a ‘standard’ Delphi approach and the study seems instead to have broadly explored the topic and refined the findings, rather than specifically achieving a pre-defined consensus on aspects of the topic.

The paper reports that participants were asked to participate in two Rounds (either Round One and Round Two, or Round Two and Round Three) and that the aim of each Round was different (Round 1 focused on identifying common points of failure in campaigns; Round 2 focused on assessing which of the points of failure were most significant; and Round 3 focused on identifying solutions). Therefore, given that the stated aim of each Round was so different, it is not clear if or how Panellists could have reconsidered previously expressed views in the subsequent Round they participated in. This also begs the question of whether Round 1 can even be considered to be a Round in the Delphi study – it may be better described as preliminary work that informed the start of a 2-Round modified Delphi study (i.e. the output of the preliminary work informed the briefing for the first Round and this constituted the major modification of the Delphi approach).

However, Delphi studies typically start with identifying the range of views expressed by the Panel on a topic and then subsequent Rounds narrow the focus in pursuit of consensus. As such, ideally all Panellists will complete all Rounds, but whilst Panellist attrition in later Rounds is common (and good quality Delphi studies typically consider the impact of this), the addition of new Panellists in subsequent Rounds is an unusual feature that seems contrary to a process that funnels consensus development through Panellists’ completion of a series of Rounds.

Moreover, the task required in Round 3 (participation in a qualitative interview) in particular does not appear to have required reconsideration of Panellists’ previously expressed views or re-rating / re-scoring them on the basis of Panel opinion(s) from Round 2. As such, it is not clear whether or what consensus was achieved – the term is not mentioned or defined in the paper (e.g., in the methods section) and no measurements / assessments of it are presented in the results or are later discussed. This appears to be fundamentally incompatible with the claim of this study being a Delphi study.

In addition to this there are other issues that need to be amended as follows:

An explanation is needed in the main text of how the descriptive statistical analysis of quantitative data from Round 2 informed the analysis presented in the results section.

The description in the results of how the Round 3 data was analysed to generate the 4 major themes is a point of method and not a result.

The results need to be presented more succinctly. Even for a qualitative analysis, the results section is too long.

The discussion section would benefit from the implications of the work being better teased out (perhaps at the end).

Reviewers' comments:

Reviewer's Responses to Questions

**Comments to the Author**

1. Is the manuscript technically sound, and do the data support the conclusions?

Reviewer #1: Yes

2. Has the statistical analysis been performed appropriately and rigorously? 

Reviewer #1: N/A

3. Have the authors made all data underlying the findings in their manuscript fully available?

Reviewer #1: Yes

4. Is the manuscript presented in an intelligible fashion and written in standard English?

Reviewer #1: Yes

5. Review Comments to the Author

Reviewer #1: Thank you for inviting me to review this paper. I confirm that I have both, subject and methodological expertise. Below are my comments and recommendations for the authors’ considerations.

ABSTRACT

- Delete “surprisingly”

- The methods section of the abstract is very short

INTRODUCTION

- Well written overall.

METHODS

- Move lines 89 to 91 up under Methods (to become line 86).

- Heading on line 88 to be rephrased “recruitment.”

- Recruitment is unclear. Provide a breakdown as to how each “sub-population” was recruited/approached.

- It is unclear what the target sample size was, and why it was chosen.

- Description of the sample usually belongs under results. See checklists like SRQR to guide the reporting of this paper.

- It is unclear as to how this paper is mixed methods (as opposed to purely qualitative).

- Would be good to see a table summarising participant characteristics.

- It would be good to see a figure illustrating the various rounds with the various stakeholders (the process by which data were collected).

- There is Figure 2 but not Figure 1. Revise/proofread.

- It is unclear how analysis was done. Was there coding? What guided the choice of themes? Were there sub-themes? Which guidelines were used? What was done to enhance trustworthiness? How many people anaylsed data? Was there cross-checking to ensure accuracy?

RESULTS and DISCUSSION

- Quite lengthy yet informative. Great to see illustrative participant excerpts in the results.

FIGURE 2 (which is probably figure 1 since there is no figure 1) is pixelated and hard to read.

FIGURE 3 (which will become figure 2) captures the results very well.

6. PLOS authors have the option to publish the peer review history of their article (what does this mean?). If published, this will include your full peer review and any attached files.

Reviewer #1: No

---

## [Author Response · Author response to Decision Letter 0]

12 Mar 2024

Please see the response to reviewers document.

---

## [Editor Report · Decision Letter 1]

22 Mar 2024

Mass media campaigns and the ‘file drawer problem’: a mixed methods study of how to avoid campaign failure

PONE-D-23-35166R1

Dear Dr. Kite,

We’re pleased to inform you that your manuscript has been judged scientifically suitable for publication and will be formally accepted for publication once it meets all outstanding technical requirements.

Kind regards,

Simon White

Academic Editor

PLOS ONE
---

## [Editor Report · Acceptance letter]

2 Apr 2024

PONE-D-23-35166R1 

PLOS ONE

Dear Dr. Kite, 

I'm pleased to inform you that your manuscript has been deemed suitable for publication in PLOS ONE. Congratulations! Your manuscript is now being handed over to our production team.

Kind regards, 

on behalf of

Professor Simon White 

Academic Editor

PLOS ONE